# Using Electronic Medical Record Data to Better Understand Obesity in Hispanic Neighborhoods in El Paso, Texas

**DOI:** 10.3390/ijerph17124591

**Published:** 2020-06-26

**Authors:** Jennifer J. Salinas, Jon Sheen, Malcolm Carlyle, Navkiran K. Shokar, Gerardo Vazquez, Daniel Murphy, Ogechika Alozie

**Affiliations:** 1Department of Molecular and Translational, Texas Tech University Health Sciences Center El Paso, El Paso, TX 79905, USA; jon.sheen@ttuhsc.edu (J.S.); macarlyl@ttuhsc.edu (M.C.); 2Department of Family and Community Medicine, Texas Tech University Health Sciences Center, El Paso, TX 79905, USA; Navkiran.shokar@ttuhsc.edu (N.K.S.); Gerardo.Vazquez@ttuhsc.edu (G.V.); Daniel.Murphy@ttuhsc.edu (D.M.); 3Sunset ID Care, El Paso, TX 79902, USA; Ogechika.Alozie@hcahealthcare.com

**Keywords:** obesity, Hispanics, socioeconomic inequities, El Paso, Texas, Texas–Mexico border, geographic weighted regression, ArcGIS, hot spot analysis, body mass index, U.S. census bureau

## Abstract

The prevalence of obesity has been persistent amongst Hispanics over the last 20 years. Socioeconomic inequities have led to delayed diagnosis and treatment of chronic medical conditions related to obesity. Factors contributing include lack of insurance and insufficient health education. It is well-documented that obesity amongst Hispanics is higher in comparison to non-Hispanics, but it is not well-understood how the socioeconomic context along with Hispanic ethnic concentration impact the prevalence of obesity within a community. Specifically studying obesity within Hispanic dominant regions of the United States, along the Texas–Mexico border will aid in understanding this relationship. El Paso, Texas is predominantly Mexican-origin Hispanic, making up 83% of the county’s total population. Through the use of electronic medical records, BMI averages along with obesity prevalence were analyzed for 161 census tracts in the El Paso County. Geographic weighted regression and Hot Spot technology were used to analyze the data. This study did identify a positive association between Hispanic ethnic concentration and obesity prevalence within the El Paso County. Median income did have a direct effect on obesity prevalence while evidence demonstrates that higher education is protective for health.

## 1. Introduction

Mexican Americans are more likely to be obese than Non-Hispanic Whites and Mexicans in Mexico [1,2]. According to recent estimates from the Centers for Disease Control and Prevention, 47% of Hispanic adults in the US are obese compared to only 37.9% of non-Hispanic whites [3]. The prevalence of obesity has been persistent and has increased over the last 20 years [4]. Of concern is the growing prevalence of related chronic disease disparities that have contributed to a lower quality of life for Mexican Americans [5].

Obesity and other Hispanic health disparities are often explained by socioeconomic inequities [6,7,8]. Imbalances in the distribution of health preserving resources delay diagnoses and treatment of disease [9]. Inadequate health preservation has been well-documented as being the result of poorly built environments, lack of insurance coverage, insufficient health literacy, and other logistical barriers [9,10]. These inequities are more likely to occur in Hispanic populations than non-Hispanic whites. Similarly, the effect of Hispanic ethnic concentration on obesity prevalence has been explained in some studies by the socioeconomic context [11,12]. However, there continues to be unanswered questions as to how the socioeconomic context and Hispanic ethnic concentration independently and collectively work to impact the community prevalence of obesity.

The relationship between Hispanic ethnic concentration and obesity has been inconsistent [13,14]. While some have found a positive correlation between the Hispanic ethnic concentration and population obesity prevalence, others have found a protective effect. [11]. These inconsistencies might be due to a number of factors related to sampling. First, there are inconsistencies within the data depending on sampling design. For example, NHANES, BRFSS, and other nationally sponsored studies are collected cross-sectionally every one or two years [15,16]. Therefore, variation in sampling, especially in regions where Hispanics represent a small proportion of the population, might lead to instability in prevalence estimates [17]. Second, sampling of different Hispanic ethnic groups might lead to variation in findings depending on the group or geographical location [18]. Finally, few studies, if any, have investigated the relationship between Hispanic ethnic concentration and obesity or other health outcomes in regions of the country, like the Southwest, where Hispanics represent the majority [12,19,20] While comparisons have been made between Hispanic-dominant and minority regions [12,19,20], little is known about the relationship between the Hispanic ethnic concentration and obesity with Hispanic dominant regions, such as the Texas–Mexico border region. This current study assessed the relationship between the Hispanic ethnic concentration and obesity prevalence in El Paso, Texas.

El Paso was a unique location for this study, as 83% of the county’s population is Mexican-origin Hispanic. An additional strength of this study was the leveraging of electronic medical record data to create BMI averages and obesity prevalence for 161 (100%) census tracts in the El Paso County. Additionally, El Paso has a diverse socioeconomic ecology, providing an opportunity to fully explore the relationship between the Hispanic ethnic concentration and socioeconomic contextual factors. This was a shortcoming of other studies that will be addressed in this study by looking at the extent to which the relationship between population Hispanic ethnic concentration and obesity prevalence is explained by population socioeconomic characteristics.

## 2. Materials and Methods

### 2.1. Electronic Medical Record (EMR) and U.S. Census Bureau Data

Data was extracted from the Electronic Medical Records (EMR) (2012–2018) systems from the Texas Tech University Health Sciences Center, El Paso Physician Clinics and University Medical Center El Paso outpatient clinics. Since both university institutions are the catchments for the county, patients represent both the insured and uninsured. Additionally, since the EMR comes from the outpatient clinics across the city, the data contributed comes from both sick, as well as, healthy visits. The data process can be seen in Figure 1. Raw data for over 2.8 million observations were cleaned and prepared for analysis. Data were first assessed for inconsistencies and duplicates. Children, or any case that did not have a valid age of 18 years or older, were removed. Errors in data entry and biometric measurements were accounted for in a series of steps. First, parameters of 14 to 100 for BMI were set and any values outside of the range were eliminated. Cases that had BMI values that seemed out of range were compared through manual calculations, before removal, when possible. The removal of children and data outside our designated parameters reduced the dataset to 665,636 observations. Repeated visits were accounted for by calculating yearly BMI averages for patients who visited the clinic more than once a year. This process reduced the dataset down to only one observation per patient/medical record number, resulting in 202,198 discrete individuals from the combined EMR.

The observations were then geocoded by their given addresses through the U.S. Census Bureau. Census tract measures were based on 2015 American Community Survey estimates [21]. We used 2015 data to correspond to the time range of the EMR data. Any non-matching addresses or non-El Paso County tracts were removed, resulting in a final total of 143,524 useable observations for geospatial analysis. Patients were clustered by census tract using their listed street address. Total number of patients per census was divided by the census tract total population to determine the proportion of patients per census tract represented in the EMR.

### 2.2. Body Mass Index (BMI) Measurement

Census tract average BMI and obesity prevalence were calculated for 161 census tracts using patient EMR data. The average BMI was therefore calculated for patients who were represented in the EMR for each respective census tract. Similarly, census tract prevalence was determined for each of the 161 census tracts by dividing the proportion of patients with a BMI of 30 or above by the total numbers of patients represented in each census tract.

#### 2.2.1. Hispanic Ethnic Concentration

Our primary explanatory variable of interest is % Hispanic, which was measured as Percentage of individuals who self-report as being Hispanic or Latino. Two potential ethnicity confounders were used as potential explanations for the effect of Hispanic ethnic concentration. Percentage (%) Immigrant—percentage of individuals who report being born in a country other than the United States in a given census tract. Percentage (%) Limited English—percentage of census tract residents whose English skills are limited or do not speak the language at all.

#### 2.2.2. Economic Confounders

Economic confounders included three measures in order to account for multiple dimensions of economic wellbeing. Percentage (%) Poverty—percentage of individuals living at or below the poverty line in a given census tract. Median Income—median census tract income. Percentage (%) Homeowner—percentage of residents who own the house that they are living in.

#### 2.2.3. Sociodemographic Confounders

Sociodemographic variables included as confounders ranged from population characteristics to health-related factors that might help explain census tract obesity in terms of behaviors or barriers to health knowledge. Total Population—total number of census tract residents. Median Age—median age of all residents of a given census tract. High School Graduation—percent of adults 25 years and older with at least a high school diploma in a given census tract. Health Insurance Coverage—percentage of adult residents who report having insurance (yes/no). Percentage (%) Walking Commuter—percentage of adult residents who walk to work (yes/no).

### 2.3. Analysis

The census tract data were assessed for normalcy of distribution and missing data. The relationship between Hispanic ethnic concentration and obesity prevalence was assessed using geographic weighted regression and hotspot analysis. Sociodemographic relationships were assessed using the same techniques, however, path analysis was also added to evaluate direct and indirect effects.

### 2.4. Geographic-Weighted Regression

Geographically weighted regression for obesity prevalence was performed to assess the unadjusted relationship between obesity, Hispanic ethnic concentration, and the potential moderating effect of socioeconomic characteristics on this relationship. An adjusted model was first run to assess the direct relationship between obesity prevalence and Hispanic ethnic concentration. A series of adjusted models were run that included individual sociodemographic characteristic variables to observe the independent effects of each on the Hispanic ethnic concentration coefficient. A final model was run to assess the collective effect of all sociodemographic variables on the relationship between obesity prevalence and Hispanic ethnic concentration. All analysis was performed using STATA ME 16 [22].

### 2.5. Hot Spot Using ArcGIS

To assess the geographic distribution of obesity prevalence in relation to key census tract socioeconomic characteristics we conducted Hotspot analysis using ArcGIS [23]. Hot Spot analysis using ArcGIS was first performed using census tract-level obesity prevalence. A z-score and associated *p*-value was generated using the Getis-Ord General G analysis. Census tract socioeconomic characteristic hotspots were also conducted to make visual comparisons by Hispanic ethnic concentration. Census tract socioeconomic characteristics were selected based on geographic-weighted regression results. Assessment was made to determine potential overlapping hotspots for the Hispanic ethnic concentration, median income, and high school graduation prevalence.

### 2.6. Path Analysis using Structural Equation Modeling

Path analysis was conducted was conducted using STATA ME 16 [24] to assess potential pathways between Hispanic ethnic concentration, median income, and high school graduation.

## 3. Results

### 3.1. Geographic-Weighted Regression Analysis

Table 1 presents the regression coefficients for the Hispanic ethnic concentration predicting the census tract obesity prevalence. Coefficients are presented for Hispanic ethnic concentration only to highlight changes as a result of the introduction of each individual variable. In the unadjusted model, for every 2.56% increase in Hispanic ethnic concentration, the average body mass index is increased by one point. In the full model, controlling for all socioeconomic variables, the coefficient is essentially reduced to zero and is no longer significant (β = 0.097, *p* = 0.863). In order to disentangle this relationship further, we conducted adjusted models controlling for each socioeconomic variable separately so that we could visualize which socioeconomic variables had the greatest modifying effect on the relationship between Hispanic ethnic concentration and obesity. Results for these analyses are also presented in Table 1. In general, the Hispanic ethnic concentration coefficient changes only slightly in each model with the exception of median income and % high school graduation. When adjusting for median income, the beta coefficient was reduced by 1.23, thereby indicating that half of the effect of the Hispanic ethnic concentration was explained by income (β = 1.33, *p* = 0.002) although the coefficient continued to be significant. Similarly, when adjusting for high school graduation, the Hispanic ethnic concentration coefficient was reduced to 1.71 (*p* = 0.001), which was again nearly half of the effect from the unadjusted model (β = 2.56).

### 3.2. Hotspots

To better understand how Hispanic ethnic concentration, obesity, median income, and high school graduation prevalence might coexist in a community that is largely Hispanic, hotspot analysis was conducted to assess for significant clusters within the El Paso County; these results are displayed in Figure 1. Hot, or significant higher averages are in red, and the cold spots are in blue. The more significant the hotspot is, the darker the color. These hotspots were generated to determine if significant overlapping patterns were present between obesity prevalence (Figure 1a), Hispanic ethnic concentration (Figure 1b), and with census tract median income (1c) and high school graduation prevalence (1d), based on the changes observed in the geographically weighted regression analysis. In all cases, statistically significant hot and cold spots were identified and can be viewed in Figure 1. Obesity had one large, significant cold spot on the westside of the county and one large hotspot on the central-eastside of the county (see Figure 1a). Similarly, there was a large cold spot in the westside and a large hotspot in the central eastside for the Hispanic ethnic concentration (see Figure 1b). Median income hotspot analysis (Figure 1c) revealed two significant hotspots—one in the west side and one in the east central areas of the El Paso County. There were also two significant cold spots. The first was located in the central area of the county and the second was located in the far east (see Figure 1c). Finally, high school graduation (see Figure 1d) yielded three hotspots—one in the westside, one in the northeast and one in the eastside. In sum, the Hispanic ethnic concentration hotspot corresponded almost directly to the obesity hotspots for the El Paso County. While there was an overlap with median income, high school graduation and obesity prevalence, there were also additional hotspots that did not correspond with what was observed for obesity. These analyses suggest that while there is a clear geographical correlation between the Hispanic ethnic concentration and obesity, the relationship was unlikely to be fully explained by significant socioeconomic indicators.

### 3.3. Path Analysis

Figure 2 displays coefficients from the path analysis. This analysis was conducted to provide better insight into the modifying effects observed in the geographically-weighted regression and hotspots. The diagram includes both indirect and direct effects for median income and % high school graduation. Hispanic ethnic concentration had a significant direct effect (1.33) on average BMI. Median income had both direct negative effects on % Hispanic ethnic concentration (−0.000004), average BMI (−0.00004), and a significant indirect effect (−0.000006). These pathways suggest that the census tract median income had a standalone effect on the census tract BMI in this Hispanic community. Additionally, despite the high overall average Hispanic ethnic concentration in El Paso, improving the income levels in census tracts with denser Hispanic prevalence might be a mechanism for reducing obesity disparities. Census tract % high school graduates, however, in this analysis only works indirectly through % Hispanic ethnic concentration. First, % high school graduation had a direct effect on % Hispanic ethnic concentration (−0.582) and an indirect effect through Hispanic ethnic concentration (−0.774). The direct effect of % high school graduate (−0.019) was not significant. In sum, in El Paso, the benefit of a higher census tract high school completion on obesity risk was only gained by offsetting the increased risk associated with higher Hispanic ethnic concentration.

## 4. Discussion

The purpose of this study was to evaluate the relationship between Hispanic ethnic concentration and obesity prevalence in El Paso, Texas, a county that is predominantly Mexican-American-origin Hispanic. The novel use of electronic medical record data enabled analysis that previous sample-based cross-sectional studies were lacking. By doing so, the results provided missing information on how contextual factors contribute to obesity risk among Hispanic populations. This study added an additional layer to the understanding of ethnic concentration as a risk factor for obesity and other health conditions by assessing how obesity was distributed within a small geographical unit (county). This study also provided an assessment of individual census tract obesity risk and risk relative to adjacent census tracts, and collectively all census tracts within the geographical unit of a county, something that was also lacking in previous studies.

### 4.1. Hispanic Ethnic Concentration is a Risk Factor for Obesity

Hispanic ethnic concentration was associated with prevalence of census tract obesity and average BMI. Analysis from the geographic-weighted regression and path analysis suggest a one point increase in average BMI for every one to two percent increase in Hispanic ethnic concentration. While El Paso is predominantly Hispanic, the hotspot analysis revealed clear and distinct overlapping hot and cold spots for both obesity and the Hispanic ethnic concentration. These findings were consistent with previous sample-based studies that have also found a significant increased prevalence of obesity in communities with higher Hispanic ethnic concentration [24,25]. Models were adjusted for both % immigrant and % limited English. However, in this study, % immigrant did not explain the effect of Hispanic ethnic concentration on obesity. Immigrant concentration was observed to provide protection from chronic diseases and the more recent immigrants tended to have better health profiles than the U.S. born [26]. Few studies have examined the effect of immigrant concentration on health outcomes of a population and evidence suggests that these relationships depend greatly on context [27]. Similar to immigrant concentration, limited English language use did not yield any significant effect on the Hispanic ethnic concentration in this study. Previous research on English language use at an individual level is mixed, dependent upon health outcome, and varies depending on the ethnic group [28,29,30,31,32,33,34].

An important question that arises with these study findings, is, if the effect of Hispanic concentration not explained by commonly used cultural variables, e.g., immigrant and language use, why is it so? One possibility, is that in ethnically homogeneous communities—New York for example—language use and immigration status serve as barriers to health information or are a characteristic of longstanding neighborhood segregation [35]. The effects of neighborhood segregation on health have been well-studied in Hispanics and other health disparate groups, primarily in the cities and metropolitan areas, where the Hispanics are the minority [36]. Language is a barrier in many of these Hispanic communities since English tends to be the only language spoken [37]. In El Paso, the majority of residents are fluent bilingual Spanish and English, thereby alleviating barriers associated with health information food access and physical activity [31]. Similarly, monolingual immigrants do not face the same challenges in El Paso in incorporating into the larger community, thereby providing a more direct pathway to assimilation than what would be experienced in other non-border communities. Therefore, the very aspects of ‘Hispanic’, which has been used traditionally to explain health and health behaviors, do not apply to El Paso and other U.S.—Mexico border at a community-level. So while the variables that are intended to measure ‘culture’ and the effect of culture on health might be measuring more complex issues at this level, like those associated with neighborhood segregation and inequality, and might be less so about the beliefs and traditions of an ethnic community.

### 4.2. Socioeconomics, Hispanic Ethnic Concentration, and Obesity

There are numerous studies that have explained the effect of Hispanic ethnic concentration as a function of socioeconomic inequality [12]. Studies that have explained the Hispanic ethnic concentration effects on obesity and other health outcomes have generally attributed these effects to socioeconomics, based on what is true of enclaves, but not of areas like the Texas–Mexico border region. Hispanic ethnic enclaves are neighborhoods or sections of larger cities or counties that are generally separated from other ethnic groups [38]. The Texas–Mexico border region is a vast area of the country that is dominated by Mexican-American Hispanics. Therefore, it is a clear contrast from Hispanic ethnic enclaves that are typically studied in large sample-based studies [39]. This study examined numerous socioeconomic census tract characteristics that could potentially explain the effect of the Hispanic ethnic concentration and found that only two studies appeared to have an effect on the relationship between Hispanic ethnic concentration and obesity [40,41].

Similar to the findings from previous studies, median income did seem to partially explain the effect of the Hispanic ethnic concentration in El Paso, Texas. However, median income also had a separate direct effect on obesity, which was demonstrated in the path analysis and potentially explained in the hotspot analysis. In the hotspot analysis, there was an overlap between obesity prevalence, Hispanic ethnic concentration, and median income. However, median income also had significant hotspots that did not overlap, suggesting that income has a very direct effect on the wellbeing of individual communities, regardless of high ethnic concentration. Measures of income are most widely used when explaining the effect of socioeconomics on racial/ethnic health disparities [42,43,44,45]. Income provides access to health preserving resources, regardless of ethnic composition of a neighborhood [46,47]. Additionally, neighborhoods with lower average incomes are often places where residents are unable to make ends meet with one job and might need multiple jobs, leaving less time for physical activity and eating healthy. Determining these specific mechanisms would provide better insight into policy making and changes that might need to be made to reduce the burden of obesity in poorer communities, which tend to be places that are also over represented with Hispanic and other health disparate groups.

Findings from this study suggest that education might be our first real clue as to the mechanism responsible for higher obesity prevalence in Hispanic communities and could be potential targets for public policy. Level of education is commonly used, like median income, to test hypotheses related to socioeconomic status. In general, there is substantial evidence that higher educational levels are protective of health. While income is more commonly used in health geography research as the primary measure of socioeconomics, education was also utilized by some, showing a consistent protective effect in population health. Few studies have shown the potential of education to explain disparities in population health by ethnic concentration [48,49,50,51,52]. High school graduation rates are historically lower in Hispanic communities compared to non-Hispanic white communities [53]. This might translate into lower health literacy in areas of El Paso that might have the highest concentrations of Hispanics. Compounded by literacy might be the variation in quality of education in areas of the county with the highest concentration of Hispanics. While education quality in ethnic communities is known to vary extensively [54], there is little effort in the health realm to understand how educational quality might translate into health outcomes [55]. This is an area of study that also needs further investigation.

### 4.3. Strengths and Limitations

This study had a number of limitations that should be noted. First, the medical record data, while extensive, only came from two health provider systems in the El Paso County, therefore the BMI data might not fully represent the true prevalence of obesity in the county. We should note, however, that in most census tracts the patient pool represented at least 5% (range 0.7% (Fort Bliss military base) to 34.9%) of the census population, which is better than what is generally sampled in national samples. On average, a census tract is about 5000, which would make the range of patients whose data contributed to this study from a given census tract to be between 250 to approximately 1745. The NHANES, the most widely used data source in this area of research, samples 5000 people per year to represent the United States population of 330 million (less than 0.0001%) [56]. The BRFSS is a cross-sectional sample of approximately 350,000, however, with a response rate of 39 to 67%; at any given year, the BRFSS sample represents between 0.0004 to 0.0007% of the United States population [57]. Even at its lowest at 0.7%, the EMR data used in this study represents a higher proportion of the population than the two most widely used data sources used for Hispanic context analysis. Hispanics that live in the El Paso are largely Mexican-American and, therefore, findings from this study might not be applicable to other Hispanic groups. Third, because the data was aggregated to a census tract level, we could not make any inferences about individual behavior, thereby making an ecological fallacy. For example, we might make the incorrect assumption that all residents of a given census tract are obese or Hispanic. Finally, to best understand health disparities, it would be optimal to be able to make comparisons with other race/ethnic groups, particularly non-Hispanic whites.

Despite the limitations, there are a number of strengths. El Paso is a unique location for this study, as 83% of the county’s population is Mexican-origin Hispanic. An additional strength of this study is the leveraging of electronic medical record data to create BMI averages and obesity prevalence for 161 (100%) census tracts in the El Paso County. El Paso has a diverse socioeconomic ecology, providing an opportunity to fully explore the relationship between Hispanic ethnic concentration and socioeconomic contextual factors. This was a shortcoming of other studies. In doing so, we were able to better identify which socioeconomic factors might be the most important for understanding the relationship between Hispanic ethnic concentration and obesity prevalence.

## 5. Conclusions

This study provided a unique perspective on the relationship between Hispanic ethnic concentration and obesity. Utilizing electronic medical records facilitated within county analysis, by taking into consideration individual census tracts in relation to other census tracts nearby or across the county. Findings from this study indicate that the Hispanic ethnic concentration does have a positive association with obesity prevalence within the El Paso County. Few socioeconomic census tract characteristics explained this relationship, with the exception of median income and high school graduation. While median income did operate indirectly through Hispanic ethnic concentration, it also had a very direct effect on obesity prevalence. High school graduation only operated through Hispanic concentration. Further studies are needed to introduce other characteristics of census tracts that might help explain the main findings from this study.

## Figures and Tables

**Figure 1 ijerph-17-04591-f001:**
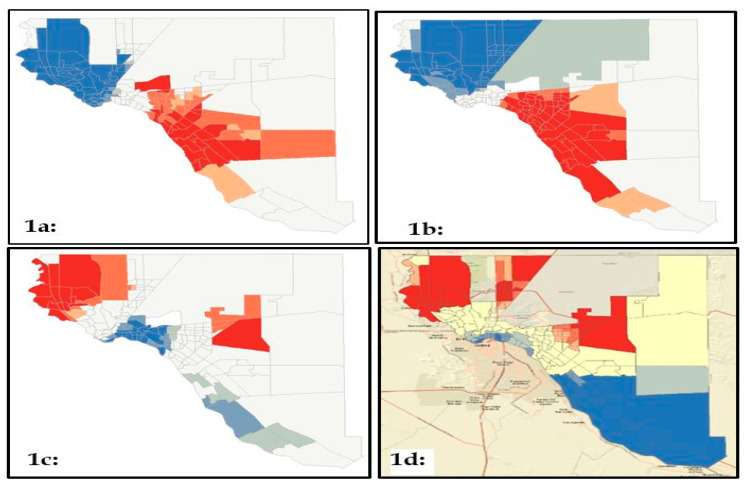
(**a**) hotspot analysis for obesity; (**b**) Hispanic ethnic concentration; (**c**) median income; and (**d**) high school graduation.

**Figure 2 ijerph-17-04591-f002:**
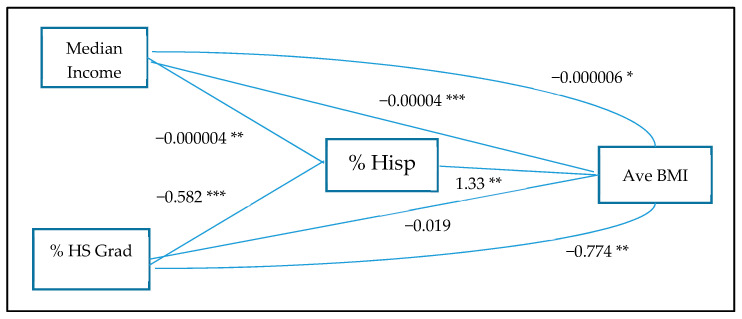
Path analysis for Hispanic ethic concentration on average BMI; * = 0.05; ** = 0.01; *** = 0.001.

**Table 1 ijerph-17-04591-t001:** Geographically weighted regression results for the census tract obesity prevalence.

Models	Coefficient	*p*-Value	Adjusted R^2^
Unadjusted	2.56	0.000	0.27
Full	0.097	0.863	0.50
Economic			
Poverty	2.30	0.000	0.27
Median Income	1.33	0.002	0.34
% Homeowner	2.57	0.000	0.26
Ethnicity			
% Immigrant	2.85	0.000	0.28
% Limited English	3.01	0.000	0.31
Demographic			
Total Population	2.53	0.000	0.27
Median Age	2.59	0.000	0.28
High School Graduation	1.71	0.001	0.29
Health Insurance Coverage	2.19	0.000	0.27
% Walking Commuter	2.48	0.000	0.27

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
