# Peer review of "Using Electronic Medical Record Data to Better Understand Obesity in Hispanic Neighborhoods in El Paso, Texas"

_ijerph, 2020, doi:10.3390/ijerph17124591_

Round 1

Reviewer 1 Report

This study investigated how socioeconomic context along with Hispanic ethnic concentration impact the prevalence of obesity within a community by using electronic medical records. This study provides a new method to investigate the risk factors of obesity and may interest readers in the field. However, the findings of the results did not provide novel information because it has been well studied that Hispanic race, lower education and lower income are risk factors for many diseases, including obesity. In addition, the figures and figure legends should be appropriately arranged and explained in the text.

Author Response

Review #1

This study investigated how socioeconomic context along with Hispanic ethnic concentration impact the prevalence of obesity within a community by using electronic medical records. This study provides a new method to investigate the risk factors of obesity and may interest readers in the field. However, the findings of the results did not provide novel information because it has been well studied that Hispanic race, lower education and lower income are risk factors for many diseases, including obesity.

Thank you for your comment. With all due respect to the reviewer, we disagree that this paper does not provide novel information. We provide rationale for this analysis in the first paragraph on the second page. The evidence has not been consistent as we point out in this paragraph that cites the inconsistencies in the literature to this point. Unfortunately, because many studies and data that are based on a sample, neighborhoods/census tracts that are ‘representative’ of Hispanic (Hispanic is not actually a racial classification, but rather an ethnic classification) communities are often also places that there is ‘lower education and lower income’- however, as we are trying to show in this paper- not all Hispanic neighbors are ‘lower education and lower income’. This is a very dangerous misconception that has gotten us nowhere in resolving persistent health disparities among Hispanic ethnic groups. As scientists, if we continue to not challenge this misconception that Hispanic ethnic concentration=lower education=lower income, we will continue perpetuate health disparities which is negligence.  We offer an alternative in this paper by looking at a community- El Paso- that is predominantly Mexican American ethnicity to disentangle this relationship. What we find, in fact, is that Hispanic does not equal lower income or lower education- as there is substantial variation in this community. We find that income operates in this community independent of Hispanic ethnic concentration, however, education very clearly works through Hispanic ethnic concentration. This is not saying that Hispanic ethnic concentration=lower education, but that in areas with the highest Hispanic ethnic concentration, education levels has an additive effect and not an explanatory effect.

In addition, the figures and figure legends should be appropriately arranged and explained in the text.

We corrected the figure 2 to make it more reader-friendly, because it seemed to be the most problematic in terms of ‘reader-friendliness’. Specifically, we reformatted the figure and added a title. We simplified the the description of the table by listing the corresponding figure next to the variable in the title (ex. Obesity (2a)) and we edited the ‘hotspot analysis’ paragraph so that the reader more easily could identify which map we were referring to in the text.

Reviewer 2 Report

This is a complicated study looking at potential correlations between Mexican Hispanic population and obesity and factors that might explain the high incidence of obesity in this population. Records of two hospitals and census data are used. 

The authors found that only median income and level of education were related to the incidence of obesity.

I am not an expert in these computer modelling of data, so I am modest in my evalaution.

My main concern is that only a limited number of data for each census was available, only of 5% of an area. Is that sufficient to reach conclusions? can the authors convice the readers better than this percentage is sufficient to reach conculsions? Could there be a bias in the data, for instance due to a high incidence of obesity in people without health coverage that did not go to the two hospitals? 

Secondly, in other studies more factors were found to be related to the incidence of obesity in this population. Is there an explanation why these factors were not found in this study? Due to the sampling method in this study or wrong conclusions in other studies?

Minor issue, in the figure it was not clear to me what the colours indicated.

Author Response

Review #2

This is a complicated study looking at potential correlations between Mexican Hispanic population and obesity and factors that might explain the high incidence of obesity in this population. Records of two hospitals and census data are used. 

The authors found that only median income and level of education were related to the incidence of obesity.

I am not an expert in these computer modelling of data, so I am modest in my evalaution.

My main concern is that only a limited number of data for each census was available, only of 5% of an area. Is that sufficient to reach conclusions? can the authors convice the readers better than this percentage is sufficient to reach conculsions?

We included substantial text to the ‘Strengths and Limitations’ section to provide readers better support for the use of our EMR data.

We should note, however, that in most census tracts the patient pool represented at least 5% (range .7% (Fort Bliss military base) to 34.9%) of the census population which is better than what is generally sampled in national samples. On average a census tract is about 5,000, which would make the range of patients whose data contributed to this study from a given census tract to be between 250 to approximately 1,745. The NHANES, the most widely used data source in this area of research, samples 5,000 people per year to represent the United States population of 330 million (less than .0001%). The BRFSS is a cross-sectional sample of approximately 350,000, however with a response rate of 39% to 67%, at any given year BRFSS sample represents between .0004% to .0007% of the United States population. Even at its lowest at .7%, the EMR data used in this study represents a higher proportion of the population than the two most widely used data sources used for Hispanic context analysis.

Could there be a bias in the data, for instance due to a high incidence of obesity in people without health coverage that did not go to the two hospitals? 

In the ‘materials and methods’ section in the paragraph describing the Electronic Medical Records, we added two sentences to clarify the origin of the data (clinics) and the payor mix (insurance and uninsured county).

“Since both university institutions are the catchments for the county, patients represent both the insured and uninsured. Additionally, since the EMR comes from outpatient clinics across the city, the data contributed comes from both sick, as well as, well visits.”

Secondly, in other studies more factors were found to be related to the incidence of obesity in this population. Is there an explanation why these factors were not found in this study? Due to the sampling method in this study or wrong conclusions in other studies?

We reformatted extensively the section titled “Hispanic ethnic concentration is a risk factor for obesity” to include more discussion on why our findings may differ from other studies. That we believe address both of these questions.

Minor issue, in the figure it was not clear to me what the colours indicated.

We added explanation of the colors to the first paragraph in the “hotspot” section in the results.

“Hot, or significant higher averages are in red, and cold spots are in blue. The more significant the hotspot is, the darker the color.”

Reviewer 3 Report

I commend the authors for their study titled,  “Using electronic medical record data to better understanding obesity disparities in Hispanic  neighborhoods in El Paso, Texas.”

General Comments

1. The study used electronic health records  and “data was extracted from the Electronic Medical Records (EMR) systems from Texas Tech University Health Sciences Center El Paso Physician Clinics and University Medical Center El Paso outpatient clinics.”  How old is the data? What reference period was used for the study? I believe so much has changed over the last one or two decades and the data and the study results may not accurately reflect the current reality because of rapidly changing demographic trends,  the shift in the racial and ethnic make-up of the population; and the shift in socio-economic structures.

2. Public health programs are mostly government-funded (often underfunded) and prevention and control measures are tailored to maximize efficiency and effectiveness of interventions through the policy-backed strategic approach of targeting all high-risk communities, not a single ethnic group; and targeting several top public health priorities, not just a single public health priority such obesity.  

3. Studies using geographical units such as census tract as sampling units or the units of data analysis are inherently characterized by analytical difficulties and can often suffer from an "ecological fallacy," which attributes collective characteristics to very dissimilar individuals or groups or the lack of agreement on the power and methods of data analysis.  

4. The study may help public health authorities to identify where the problem of obesity IS (or more appropriately where the problem WAS  5, 10 or 20 years ago if the data is as old ) but does not address the disparities. in the prevailing social determinants of obesity and related disparities in Hispanic communities. Despite the title, there are no new findings or discussion of disparities, or comparison with other ethnic or racial groups in this study. The study tries to explore an association between obesity as assessed by the measurement of BMI and some sociodemographic variables in Hispanics.

 5. It is not as informative as we wish it to be to display beta coefficients on a bar graph. Instead, the bar graph could have been more useful if it displayed the disparities among different ethnic and racial groups in the stated parameters: poverty, income, homeownership, immigration status, health insurance, etc.  The Beta Coefficients could be displayed in a table or some other chart.

  6.  In sum, despite the apparent rigor of the statistical methods used to analyze data in this study, the usefulness of the study findings is eclipsed by (1) the inherent ecological fallacy resulting from the study design and nature of the sampling unit; the sample consists of patient population which is not necessarily a representative of obesity patterns in the general population. (2)  the failure to investigate the actual determinants of the disparities among different ethnic and racial groups in public health problem in question, i.e., obesity.   

7. I think the usefulness of the research findings in this study as it stands currently to influence changes in public health policy and intervention programs is limited.  

Author Response

Review #3

I commend the authors for their study titled,  “Using electronic medical record data to better understanding obesity disparities in Hispanic  neighborhoods in El Paso, Texas.”

General Comments

1. The study used electronic health records  and “data was extracted from the Electronic Medical Records (EMR) systems from Texas Tech University Health Sciences Center El Paso Physician Clinics and University Medical Center El Paso outpatient clinics.”  How old is the data? What reference period was used for the study?

We added the date range 2012-2018 to the data description in the methods section.

I believe so much has changed over the last one or two decades and the data and the study results may not accurately reflect the current reality because of rapidly changing demographic trends,  the shift in the racial and ethnic make-up of the population; and the shift in socio-economic structures.

We do agree that there are substantial shifts in demographics over the course of decades, for that reason we used census data from roughly the same time period. We added a statement to that effect in the second paragraph, “We used 2015 data to correspond to the time range of the EMR data.”

  1. Public health programs are mostly government-funded (often underfunded) and prevention and control measures are tailored to maximize efficiency and effectiveness of interventions through the policy-backed strategic approach of targeting all high-risk communities, not a single ethnic group; and targeting several top public health priorities, not just a single public health priority such obesity.  

We agree with this statement. Our intention was not to minimize the challenges other health disparate groups face, but to better understand the nature of health disparities by race/ethnic group by conducting a ‘what if’ exercise on how relationship might look in a community where a health disparate group- Mexican Americans- were the majority and race/ethnic inequality within a city geography was not as defined as in a larger, more homogeneous community.

3. Studies using geographical units such as census tract as sampling units or the units of data analysis are inherently characterized by analytical difficulties and can often suffer from an "ecological fallacy," which attributes collective characteristics to very dissimilar individuals or groups or the lack of agreement on the power and methods of data analysis.  

We agree with this reviewer’s observation. We modified the second to last sentence in the ‘strengths and limitations’ paragraph to specifically mention ecological fallacies- which was not included in the previous draft. “Third, because the data is aggregated to a census tract level, we cannot make any inferences on individual behavior thereby making an ecological fallacy.”

Reviewer 4 Report

Thank you for submitting this interesting analysis. It provides important insight for researchers to better understand the causes of obesity among Hispanic Americans of Mexican descent. That said, there are several issues with this manuscript that need to be addressed prior to acceptance.

From the beginning, it is evident that the manuscript needs to be proofread more carefully. In the title, “understanding” should be “understand”. In the abstract, on line 21, ‘countries” should be “country’s”. Additionally, on page 6, line 286, the sentence before [58] is incomplete. These are just some of the examples of careless mistakes. Beyond this, there are several points that require clarification throughout the manuscript.

Introduction

On line 32, I’m assuming that “…47.0% of Hispanic adults…” refers to those in the US. While it might be redundant, because you speak about Mexicans in Mexico in the previous sentence, it would be helpful to identify which group you’re referring to in the second sentence.

Line 35/36: it’s redundant to say “Mexican Americans” in the “United States”.

Line 54: add comma after, “if any”.

Line 61-65, this does not belong in the intro. It should be incorporated into the discussion.

Materials and Methods

You refer to Figure 1; this does not match what is shown in Figure 1.

Line 95: % Hispanic does not belong here.

Line 117: delete the word “for” before direct.

Geographic Weighted Regression

You name and discuss analysis for some (sociodemographic), but not all, confounders. Overall, labeling of the different confounders needs to be updated and streamlined.

Results

Results is actually “3”. It looks like it is it’s currently listed as a subheading of ‘materials and methods’.

Interpretation of analysis is unclear. As reported in Figure 1, it appears that some factors may need to be reverse coded. For instance, as shown, for every 1.72% increase in high school graduation concentration, average BMI is increased (by 1 point). Is this the case or should this read 1.72% decrease in high school graduation concentration, average BMI is increased? Similarly, more insurance increases likelihood of obesity?

Hotspots

This area is unclear for a variety of reasons. It is very challenging to decipher the differences in the hotspots as you described given a) the size of the figures; b) lack of a legend accompanying the figures; and c) difficulty deciphering between hotspots in their current color scheme (you’d be better off using color or texture to distinguish differing concentrations).

Path Analysis

Line 200: For clarity, add “on average BMI” after significant indirect effect.

Discussion (now section 4)

Line :228 – “within”

Overall, directionality appears to need further explanation. The language throughout the discussion is rather vague. Further, if as discussed in Line 276, education is protective of health, wouldn’t that show that higher education is associated with lower BMI?

Author Response

Review #4

Thank you for submitting this interesting analysis. It provides important insight for researchers to better understand the causes of obesity among Hispanic Americans of Mexican descent. That said, there are several issues with this manuscript that need to be addressed prior to acceptance.

From the beginning, it is evident that the manuscript needs to be proofread more carefully. In the title, “understanding” should be “understand”.

We made this change

In the abstract, on line 21, ‘countries” should be “country’s”.

We made this change

Additionally, on page 6, line 286, the sentence before [58] is incomplete.

We made this change

These are just some of the examples of careless mistakes. Beyond this, there are several points that require clarification throughout the manuscript.

Introduction

On line 32, I’m assuming that “…47.0% of Hispanic adults…” refers to those in the US. While it might be redundant, because you speak about Mexicans in Mexico in the previous sentence, it would be helpful to identify which group you’re referring to in the second sentence.

We made this change

Line 35/36: it’s redundant to say “Mexican Americans” in the “United States”.

We made this change

Line 54: add comma after, “if any”.

We made this change

Line 61-65, this does not belong in the intro. It should be incorporated into the discussion.

We moved this section to strength and limitations.

Materials and Methods

You refer to Figure 1; this does not match what is shown in Figure 1.

We removed the incorrect table and updated the figures.

Line 95: % Hispanic does not belong here.

We updated section

Line 117: delete the word “for” before direct.

 We made his change.

Geographic Weighted Regression

You name and discuss analysis for some (sociodemographic), but not all, confounders. Overall, labeling of the different confounders needs to be updated and streamlined.

Results

Results is actually “3”. It looks like it is it’s currently listed as a subheading of ‘materials and methods’.

We made that change.

Interpretation of analysis is unclear. As reported in Figure 1, it appears that some factors may need to be reverse coded. For instance, as shown, for every 1.72% increase in high school graduation concentration, average BMI is increased (by 1 point). Is this the case or should this read 1.72% decrease in high school graduation concentration, average BMI is increased? Similarly, more insurance increases likelihood of obesity?

We made extensive edits to the results section and believe that the main take home points are clearer. This includes switching out Figure 1 for a table (now Table 1) and better description of the significant patterns in the coefficients in that table. We also, provided a better link between the table, hotspot analysis and path analysis.

Hotspots

This area is unclear for a variety of reasons. It is very challenging to decipher the differences in the hotspots as you described given a) the size of the figures; b) lack of a legend accompanying the figures; and c) difficulty deciphering between hotspots in their current color scheme (you’d be better off using color or texture to distinguish differing concentrations).

In response to this and other reviewer comments we edited this section extensively.

Path Analysis

Line 200: For clarity, add “on average BMI” after significant indirect effect.

We made this change.

Discussion (now section 4)

Line :228 – “within”

We made this change.

Round 2

Reviewer 3 Report

The authors have addressed most of this reviewer's questions and comments. Items that are not addressed such as possible bias resulting from an ecological fallacy associated with the design of this study should be mentioned as limitations in the Limitations Section. Further review of the manuscript is left to the discretion of the authors. 

Author Response

The authors have addressed most of this reviewer's questions and comments. Items that are not addressed such as possible bias resulting from an ecological fallacy associated with the design of this study should be mentioned as limitations in the Limitations Section. Further review of the manuscript is left to the discretion of the authors.

Response: Ecological fallacies of the study are expounded to address potential bias on Line 316 of the Limitations section. We added an additional sentence to further address how ecological fallacies could be made from this study.

Reviewer 4 Report

The authors have addressed the comments made on initial review.

The final paragraph from the original introduction is still present, though it has also been moved to the strengths/limitations. Make sure that this change is incorporated into/reflected in the final version of the paper. Further, given the new placement of this paragraph, the final sentence is now misplaced. Please make sure to proofread before submitting the final version of this manuscript.

Author Response

The final paragraph from the original introduction is still present, though it has also been moved to the strengths/limitations. Make sure that this change is incorporated into/reflected in the final version of the paper. Further, given the new placement of this paragraph, the final sentence is now misplaced. Please make sure to proofread before submitting the final version of this manuscript.

Response: Typographical errors corrected on Line 206 and 311.

Revised the final sentence to better fit its location and relevance in the Strengths section at Line 324.